# Immune Checkpoint Profiling in Humanized Breast Cancer Mice Revealed Cell-Specific LAG-3/PD-1/TIM-3 Co-Expression and Elevated PD-1/TIM-3 Secretion

**DOI:** 10.3390/cancers15092615

**Published:** 2023-05-04

**Authors:** Christina Bruss, Kerstin Kellner, Veruschka Albert, James A. Hutchinson, Stephan Seitz, Olaf Ortmann, Gero Brockhoff, Anja K. Wege

**Affiliations:** 1Department of Gynecology and Obstetrics, University Medical Center Regensburg, 93053 Regensburg, Germany; christina.bruss@ukr.de (C.B.); sseitz@csj.de (S.S.); oortmann@csj.de (O.O.); gero.brockhoff@ukr.de (G.B.); 2Department of Surgery, University Hospital Regensburg, 93053 Regensburg, Germany; james.hutchinson@ukr.de

**Keywords:** humanized tumor mice (HTM), breast cancer, hematopoietic stem cells (HSC), TIM-3, LAG-3, galectin-9, PD-1, PD-L1, soluble checkpoint, immunotherapy

## Abstract

**Simple Summary:**

Different immunotherapies have been approved for the treatment of a multiplicity of cancers. However, a large proportion of patients do not respond or develop resistance, meaning that specified treatment combinations are required to enhance individual therapy efficiencies. A combined anti-PD-1/anti-LAG-3 therapy has already been approved for the treatment of melanoma patients. Here, we describe the checkpoint expression patterns and secretion of, e.g., TIM-3, LAG-3, galectin-9 and PD-(L)1/2 in breast cancer-specific humanized tumor mouse models. We quantitatively determine the breast cancer subtype-specific checkpoint co-expression and release. These data profoundly demonstrate the potential of humanized tumor mice as a significant mainstay for preclinical immunotherapeutic trials.

**Abstract:**

Checkpoint blockade is particularly based on PD-1/PD-L1-inhibiting antibodies. However, an efficient immunological tumor defense can be blocked not only by PD-(L)1 but also by the presence of additional immune checkpoint molecules. Here, we investigated the co-expression of several immune checkpoint proteins and the soluble forms thereof (e.g., PD-1, TIM-3, LAG-3, PD-L1, PD-L2 and others) in humanized tumor mice (HTM) simultaneously harboring cell line-derived (JIMT-1, MDA-MB-231, MCF-7) or patient-derived breast cancer and a functional human immune system. We identified tumor-infiltrating T cells with a triple-positive PD-1, LAG-3 and TIM-3 phenotype. While PD-1 expression was increased in both the CD4 and CD8 T cells, TIM-3 was found to be upregulated particularly in the cytotoxic T cells in the MDA-MB-231-based HTM model. High levels of soluble TIM-3 and galectin-9 (a TIM-3 ligand) were detected in the serum. Surprisingly, soluble PD-L2, but only low levels of sPD-L1, were found in mice harboring PD-L1-positive tumors. Analysis of a dataset containing 3039 primary breast cancer samples on the R2 Genomics Analysis Platform revealed increased TIM-3, galectin-9 and LAG-3 expression, not only in triple-negative breast cancer but also in the HER2^+^ and hormone receptor-positive breast cancer subtypes. These data indicate that LAG-3 and TIM-3 represent additional key molecules within the breast cancer anti-immunity landscape.

## 1. Introduction

Checkpoint inhibitors have revolutionized the treatment options for different types of cancers. In general, the inhibitory receptors on immune cells are responsible for protection against autoimmunity and block chronic inflammation to prevent tissue damage. However, tumors take advantage of this checkpoint mechanism via the upregulation of the ligands for the shutdown of immune cell function. Targeting these receptors enables the “release of the brake” and activates tumor-specific immune cell responses against cancer. The first checkpoint inhibitor against cytotoxic T lymphocyte protein 4 (CTLA-4, CD152, ipilimumab; Bristol-Myers Squibb; interacts with CD80 (B7.1), alternatively with CD86 (B7.2)) has been approved in March 2011 in the US for the treatment of patients with unresectable or metastatic melanoma [1]. In 2014, the monoclonal antibody (mab) pembrolizumab (anti-programmed cell death protein 1, anti-PD-1, Merck) has been made available for the treatment of patients with disease progression of ipilimumab-refractory melanoma [2]. Based on data from Impassion130 (NCT02425891), atezolizumab (anti-programmed cell death 1 ligand 1, anti-PD-L1 mab, Roche) has been approved by the FDA in March 2019 for the treatment of advanced or metastatic triple-negative breast cancer (TNBC) patients with tumor-infiltrating immune cells expressing PD-L1 (≥1%, tumor area) [3]. However, despite the achievements of the CTLA-4/PD-1/PD-L1 blockade strategies, many patients still do not respond to these drugs or even show hyperprogression. Therefore, new attempts are ongoing to define other surface molecules targeting the inhibitory pathway, including the lymphocyte activation gene 3 (LAG-3; CD223) and T cell immunoglobulin and mucin-domain-containing 3 protein (TIM-3). LAG-3 is a member of the immunoglobulin superfamily and is typically expressed on activated T, B and natural killer (NK) cells as well as on plasmacytoid dendritic cells (DC). Similar to the CD4 molecule on T cells, LAG-3 binds to the major histocompatibility complex class II (MHC II) molecules, but with an up to 100-fold higher affinity than CD4. By preventing the interaction of CD4 and the T cell receptor (TCR) with MHC II, the intracellular activation pathway signaling is diminished [4]. Other possible ligands of LAG-3 include liver endothelial cell lectin (LSECtin), fibrinogen-like protein 1 (FGL1), galectin-3 (Gal-3), and alpha-synuclein fibrils, which have been reported to block activation and cytokine secretion in immune cells [4]. Blockade of LAG-3 in peripheral blood mononuclear cells isolated from CLL patients resulted in enhanced CD4 and CD8 T cell proliferation with elevated secretion of different cytokines, such as interferon-γ, tumor-necrosis-factor-α, and interleukin-2 [5]. Moreover, LAG-3 expressed on regulatory T cells (Tregs) blocked maturation of DCs via MHC II cross-linking [6]. Moreover, the co-expression of CD49b and LAG-3 has been identified as a marker of CD4 type 1 regulatory T cells in mice and human [7]. Interestingly, LAG-3 and PD-1 have been frequently found to be co-expressed, which is considered to exhibit synergistic immunosuppression [8]. Subsequently, Woo and colleagues showed beneficial effects in relation to tumor growth inhibition in a colon adenocarcinoma mouse model when both molecules were targeted [9]. TIM-3 is another checkpoint expressed on T cells, macrophages, and DCs, and it binds galectin-9 but also high-mobility group box 1 protein (HMGB1), phosphatidylserine (PtdSer), and carcinoembryonic antigen-associated cell adhesion molecules (Ceacam-1) [10]. HMGB1 has been found to be secreted from tumor cells, which also mediated anti-tumor immune responses via activating Toll-like receptor 2 + 4 signaling [11,12]. Galectin-9 has been detected on a variety of cancers and its secretion from human tumor cell lines can be induced by T lymphocytes, attenuating their function [13]. However, galectin-9 can also interact with the stimulatory molecules CD137 and CD40 on T cells. TIM-3 is co-expressed with PD-1 on exhausted T cells, and TIM-3 signaling, as an escape mechanism, is associated with resistance to anti-PD-1 therapy [14]. Dual blockade of TIM-3 and PD-1 induces tumor regression [15]. In addition to PD-L1, programmed cell death 1 ligand 2 (PD-L2) is another corresponding partner of PD-1 and is expressed on tumor cells and antigen presenting cells. It contributes to cancer progression and immune escape mechanisms, which might be a possible explanation for successful anti-PD-1 therapies in patients with PD-L1-negative tumors. Other immune-modulatory antibodies, e.g., agonistic antibodies for activating molecules such as CD137 (4-1BB), CD40, CD27 or glucocorticoid-induced TNFR-related protein (GITR), are under investigation or have already entered clinical trials.

Here, we quantitatively analyzed the expression and the secretion of different checkpoint molecules in humanized tumor mice (HTM) models specific for HER2^+^, hormone receptor positive, and TNBC. HTM develop a human immune system and human tumor growth [16], and they have already enabled the identification of novel therapeutic antibodies [17], prognostic and potentially predictive markers [18], and probably side effects during immunomodulatory cytokine stimulation [19]. The expression of different checkpoint molecules has already been described in patient-derived (humanized) tumor mice (PDX) mice using multi-color-flow cytometry [20]. Here, we evaluated the expression and the secretion of different human immune checkpoints in HTM and human tissues. Furthermore, we assessed the utility of HTM in relation to investigating predictive or prognostic markers during checkpoint inhibitor combination treatments.

## 2. Materials and Methods

### 2.1. Breast Cancer Cell Lines

The breast cancer cell lines JIMT-1 (DSMZ number ACC-589), MCF-7 (ATCC number HTB-22) and MDA-MB-231 (ATCC number HTB-26) were obtained from the American Type Culture Collection (ATCC, LGC Standards, Wesel, Germany). The cells were cultured in RPMI 1640 (JIMT-1; Life Technologies, Carlsbad, CA, USA, Cat. No. 52400041) or DMEM (MCF-7 and MDA-MB-231; Life Technologies, Cat. No. 31885049) supplemented with 5% fetal calf serum (FCS) (Gibco, New York, NY, USA, 10270106) under standard cell culture conditions.

### 2.2. Isolation of Human CD34^+^ Hematopoietic Stem Cells from Umbilical Cord Blood

Human CD34^+^ hematopoietic stem cells were isolated from cord blood as described before [16]. Briefly, the two-step procedure included the isolation of mononuclear cells via density gradient centrifugation followed by enrichment of the CD34^+^ cells with immunomagnetic beads (Miltenyi Biotech, Bergisch Gladbach, Germany) according to the manufacturer’s protocol. The viability and purity of the obtained cell suspension/fraction were assessed with flow cytometry. The CD34^+^ cells were cryopreserved in liquid nitrogen until transplantation.

### 2.3. Generation of Humanized Tumor Mice

Humanized tumor mice were generated as previously described [16]. In brief, NOD.Cg-Prkdcscid Il2rgtm1Wjl/SzJ (NSG) mice (Charles River) were bred and housed in a specialized pathogen-free facility at the University of Regensburg. For humanization, the newborn pups were irradiated with a sub-lethal dose of 1 Gy, and 3 h later, 1 × 10^5^ CD34^+^ cells were injected intra-hepatically. After nine weeks, blood was collected via the lateral saphenous vein and the state of the reconstituted human immune system was controlled. One week later, the mice were transplanted under anesthesia (5 mg/kg midazolam, 0.05 mg/kg fentanyl and 0.5 mg/kg medetomidine). The transplant was inserted orthotopically in the mammary fat pad and consisted either of tumor cells or patient-derived tumor tissue. The primary tumor used for the PDX mice was classified as ER^+^, HER2^+^, and PR^−^ by a pathologist. Anesthesia was antagonized using flumazenil (0.5 mg/kg), atipamezol (2.5 mg/kg) and naloxon (1.2 mg/kg).

### 2.4. Ethics Statements

The cord blood and patient samples were taken with approval from the Ethics Committee of the University of Regensburg (permission no. 18-1039-101). All the participants provided written informed consent. The animal work was approved based on European guidelines and the national regulations of the German Animal Protection Act by the local veterinary authorities of the district government (permission no. RUF 55.2-2532.2-803-16).

### 2.5. Flow Cytometry

Expression of membrane-bound proteins on the tumor and immune cells was analyzed via flow cytometry with a FACSCanto-II (BD Biosciences, San Jose, CA, USA), which was run by the Diva software ver. 7.0 (BD Biosciences, San Jose, CA, USA).

Prior to analysis, the solid tissue samples were mechanically dissociated into single-cell suspensions using 40 µm cell strainers. The following antibodies were used for staining, which were purchased from BD Biosciences: αCD3-FITC (555332, UCHT1), αCD3-PerCP (332771, SK7), αCD4-APC-H7 (641398, SK3), αCD45-APC (555485, HI30), and αMHCII-BB700 (742224, Tu39). The following antibodies were purchased from BioLegend (San Diego, CA, USA): αCD8a-BV510 (301048, RPA-T8), αCD44-BV510 (103044, IM7), αCD45-PerCP-Cy5.5 (304010, HI30), αCD45RA-BV421 (304130, HI100), αCD137-PeCy7 (309818, 4B4-1), αEpCAM-AF647 (324212, 9C4), αICAM-AF647 (353114, HA58), αHer2-AF488 (324410, 24D2), αPD-1-AF647 (329910, EH12.2H7), αTIM-3-BV421 (345008, F38-2E2), and αPD-L1-BV21 (329714, 29E2A3). The following antibodies were used for staining and purchased from eBiosciences (San Jose, CA, USA): αCD24-PeCy7 (25-0247-42, eBioSN3 SN3 A5-2H10), αCD27-PeCy7 (25-0279-42, O323), and αMHCI-Pe (MA1-10346, MEM-123). The antibody αLAG-3-Pe (FAB2319P, polyclonal goat IgG) was ordered from R&D Systems and αCD49b-FITC (IM1425, Gi9) from Beckman Coulter (Brea, CA, USA).

Appropriate mouse immunoglobulin antibodies were used as isotype controls for the antigen-specific staining. The results, including the t-SNE analysis, were analyzed using the FlowJo software v10.8 (BD Biosciences, San Jose, CA, USA). For the t-SNE analysis, the CD4 and CD8 T cells were extracted by means of gating (single CD45, CD3, CD4 and CD8) and were transferred to new files. Analysis of the clustering was performed based on the PD-1, LAG-3 and TIM-3 expression.

### 2.6. BioLegend’s LEGENDplex™ Bead-Based Immunoassay

The soluble molecules in the serum of the HTM and in the cell culture supernatants were quantified using the LEGENDplex™ HU Immune Checkpoint Panel 1 (Cat. No. 740867, analyzed molecules: sCD25, 4-1BB, sCD27, B7.2, free active TGF-ß1, CTLA-4, PD-L1, PD-L2, PD-1, TIM-3, LAG-3, and Galectin-9) and the LEGENDplex™ HU Essential Immune Response Panel (Cat. No. 740929, analyzed molecules: IL-4, CXCL10, IL-1β, TNF-α, CCL2, IL-6, CXCL8, and free active TGF-β1) according to the manufacturer’s protocol. The data were processed with the LEGENDplex™ Data Analysis Software Suite.

### 2.7. ELISA

The soluble HMGB1 and C-X3-C motif chemokine ligand 1 (CX3CL1) in the serum of the HTM and in the cell culture supernatants were assessed using the HMGB1 Express ELISA Kit (Tecan, Cat. No. 30164033) and the CX3CL1/Fractalkine DuoSet ELISA (R&D, Cat. No. DY365) according to the manufacturer’s protocol and using the plate reader from CLARIOstar (BMG Labtech, Ortenburg, Germany).

### 2.8. Database Analysis

The gene expression, Kaplan–Meier analysis and correlation of *PDCD1*, *LAG3* and *HAVCR2* among each other as well as *HAVCR3* with *LGALS9* were generated with the R2: Genomics Analysis and Visualization Platform (http://r2.amc.nl, accessed on 23 January 2023) using the “R2: Tumor Breast (primary)−Gruvberger-Saal−3207−tpm−gse202203” dataset, which was firstly published by Dalal and colleagues [21]. Absolute r values between 0 and 0.25 were regarded as not correlated, 0.25 and 0.50 as weak, 0.50 and 0.75 as moderate, and 0.75 and 1.00 as very strong correlation.

For the Kaplan–Meier estimation curves, the patients were divided into a high or low expressing group via the median expression in the TNBC patient group. The cut-off value of the gene expression level defined in the TNBC group was applied for all the subtypes.

### 2.9. Statistical Analyses

The results are shown either as median or mean and standard deviation (SD), as described in the figure legends. Statistical and correlation analyses was performed using the GraphPad Prism software (version 6, La Jolla, CA, USA). Data are judged to be statistically significant when *p* < 0.05 according to the one-way ANOVA and Tukey’s multiple comparisons test. In the figures, asterisks denote statistical significance (* *p* < 0.05, ** *p* < 0.01, *** *p* < 0.001). The correlations were calculated using a Pearson test. Absolute r values between 0 and 0.25 were regarded as not correlated, 0.25 and 0.50 as weak, 0.50 and 0.75 as moderate, and 0.75 and 1.00 as very strong correlation.

## 3. Results

### 3.1. Colonization of Human Immune Cells upon HSC Transplantation

A high percentage of human CD45^+^ immune cells were generated in the HSC transplanted NSG at the age of 9 weeks (mean 49.4% ± 17.1 SD) in all the mice, with a slightly different percentage in the different HTM models (Appendix A). However, the percentage of reconstituted human CD45^+^ cells in the spleen of all the HTM at the end of the experiment was very similar, with an overall reconstitution level of 62.9% ± 15.3. As expected, the percentage of human T cells (CD3^+^) at the age of nine weeks was low, but it increased in the MDA-MB-231 and JIMT-1 transplanted HTM to equivalent levels with the B cell population at the age of 22 weeks in the spleen (Appendix A). In the MCF-7 transplanted HTM, when analyzed at the age of 15 weeks, the T cells population just increased to 9.9% ± 7.5. The myeloid (CD33^+^) population was low in the peripheral blood (1.8% ± 1.4) and just slightly more in the spleen (2.1% ± 1.8) at the end of the experiments (Appendix A).

### 3.2. Differences in Phenotype of Breast Cancer Cell Lines from Different Subtypes, Which Might Contribute to Immunogenicity

After the successful engraftment of the human immune system, the mice were transplanted with the three breast cancer cell lines, JIMT-1 (HER2^+^/ER^−^), MDA-MB-231 (HER2^−^/ER^−^, TNBC) and MCF-7 (HER2^−^/ER^+^), mimicking different breast cancer subtypes. As the phenotype of the tumor cells might affect the immune cell infiltration into the tumor, we characterized the cell lines regarding their MHC I and MHC II expression, which plays a pivotal role in initiating and mediating immune responses by antigen presentation.

All three cell lines were positive for MHC I (Figure 1A,B), which interacts mainly with CD8 as a co-receptor. However, MDA-MB-231 was characterized by a lower mean fluorescence intensity for MHC I (Figure 1C). By contrast, MHC II, commonly expressed by professional antigen-presenting cells, was highly expressed on MDA-MB-231, followed by the JIMT-1 cells, while the MCF-7 cells showed only a low expression (Figure 1A–C). Immunohistochemical staining confirmed these observations. With respect to the immune checkpoint therapies, the cell lines were tested for their PD-L1 expression. In line with previous results [22], HER2^+^ JIMT-1 showed the highest PD-L1 expression, followed by TNBC MDA-MB-231. MCF-7 did only express low or no levels of PD-L1 (Figure 1A–C). Interestingly, the FISH analysis revealed that the PD-L1 gene copy number in the MCF-7 and JIMT-1 cells was almost unaltered (i.e., nearly two chromosomes 9 and two gene copies per cell), while the MDA-MB-231 cells showed a loss of the *PDCD1LG2* gene. With special regard to a tumor stem cell-like phenotype (CD24^low^, CD44^high^) and the potential to metastasize (represented by a CD44^+^ phenotype), CD24 and CD44 were included in the analysis. JIMT-1 appeared double-positive for both markers, whereas MDA-MB-231 was CD24^−^ but CD44^+^, inverse to the pattern observed on the MCF-7 cells (Figure 1A–C).

As inflammation and cytokine production are not only indicative of an effective immune response but also proposed to initiate metastasis and tumor promotion, amongst other phenomena, we analyzed several cytokines, ligands for chemokines and other soluble factors known to be important in the breast cancer microenvironment (Table 1). Analysis of the soluble molecules secreted into the cell culture supernatants revealed that in particular MDA-MB-231, in contrast to JIMT-1 and MCF-7, was characterized by the capacity to release several factors. IL-6, CXCL8, and CX3CL1, which are known as proinflammatory or chemoattractive molecules, were highly elevated. Interestingly, the PD-L1-expressing JIMT-1 and MDA-MB-231 cells did not produce soluble PD-L1 but rather sPD-L2. Next to these factors, HMGB1 was secreted in high amounts not only by MDA-MB-231 but also by JIMT-1 and MCF-7. In addition, MDA-MB-231 was characterized by the release of galectin-9. HMGB1 and galectin-9 can activate the checkpoint protein TIM-3, which might lead to T cell anergy.

### 3.3. Checkpoint Therapy-Related Soluble Proteins Are Elevated in the Serum of Humanized Mice Transplanted with Breast Cancer

Upon activation, T cells express checkpoint proteins such as PD-1, TIM-3 or LAG-3, which can limit the immune response when interacting with their ligands. Several immune checkpoint proteins as well as their corresponding ligands can be found in their soluble variant in the blood of patients, contributing to the activation or inhibition of the immune response. To test whether these soluble factors can be found in humanized mouse models, we analyzed the serum of mice transplanted with JIMT-1, MDA-MB-231, MCF-7 cells or patient-derived tissue using a bead-based immunoassay (Figure 2). In order to exclude soluble factors derived from the NSG background, we also analyzed the serum of non-transplanted mice as a negative control. No soluble factors were detectable in these samples, indicating that any measured soluble checkpoint protein must be secreted either by the human immune system or by the transplanted human tumor cells.

We quantified the soluble proteins related to T cell activation (sCD25, sCD27, 4-1BB), soluble checkpoint proteins (CTLA-4, sPD-1, sTIM-3 and sLAG-3) and ligands thereof (B7.2, sPD-L1, sPD-L2, and galectin-9) in the HTM and PDX serum (Figure 2A). In particular, in the JIMT-1 and MDA-MB-231 mice, the T cell activation-related molecules were increased in comparison to the MCF-7 or PDX mice, indicative of an effective T cell dependent immune response. The soluble CTLA-4 levels were rather low, while the amounts of sPD-1 and sPD-L1 were found to be only marginally elevated. However, high sPD-L2 concentrations were found in the serum of the JIMT-1, MDA-MB-231 and PDX mice but not in the MCF-7 mice. Interestingly, sTIM-3 and galectin-9 were detectable in higher concentrations in all the mice, but in particular, in the MDA-MB-231 group. Analysis of sLAG-3 indicated a subordinated role for this compound in the serum of the HTM mice.

Some mice transplanted with MDA-MB-231 cells partially rejected tumor engraftment (Figure 2A), which is believed to be the consequence of recognition of discrete tumor antigens by the host’s immune system. Analysis of the soluble factors in the serum of these mice revealed slightly to moderately reduced levels in comparison to the tumor-bearing mice (Figure 2A). sPD-L2 was strongly elevated in the mice with engrafted tumor growth, indicative of the release of this molecule by the tumor microenvironment. Moreover, the sTIM-3 levels were decreased in these mice.

The membrane-bound PD-L1 on the tumor cells was slightly correlated with the secretion of sPD-1 and sPD-L1 in the serum (Figure 2B), whereas no correlation between the membrane-bound PD-L1 on the splenocytes and sPD-1 or sPD-L1 was observed (Figure 2C). No relation between the membrane-bound PD-L1, neither on the tumor cells nor on the splenocytes, was found with sPD-L2 (Figure 2B,C). The membrane-bound MHC II (a possible ligand of LAG-3) on the tumor cells did not correlate with sLAG-3 (Figure 2B). The membrane-bound PD-1 on the splenocytes was correlated with sPD-1 but not with sPD-L1 or sPD-L2 (Figure 2C). To exclude the influence of the tumor size, the correlation between the soluble factors and tumor size was calculated and no correlations were found.

As combination therapies in the field of immunotherapeutic approaches are discussed, we correlated the soluble forms of sPD-1, sLAG-3 and sTIM-3 with each other and found a weak positive relation between sPD-1 and sLAG-3 (Figure 2D). Moreover, a moderate correlation between galectin-9 and sTIM-3 was observed.

### 3.4. CD4 T Cells Are Characterized by Elevated LAG-3, CD8 T Cells and by Elevated TIM-3 Expression in the Tumor Microenvironment of HTMs

Next, we analyzed whether the level of soluble checkpoints in the serum was related to the membrane-bound molecules on the T cells. The soluble and membrane-bound forms were determined in the blood and several organs (blood, spleens, lymph nodes, thymi and tumors) in the MDA-MB-231 and PDX transplanted mice. We found a moderate correlation between sTIM-3 and TIM-3 expression on the CD4 (r = 0.6816) and CD8 T cells (r = 0.5844) but no relation between LAG-3/sLAG-3 or PD-1/sPD-1.

Even though there was no strong evidence of correlations, the tumor-infiltrating CD4 and CD8 T cells were characterized by high PD-1 expression in both mice groups (Figure 3), whereas in the PDX mice, the overall immune cell infiltration was low in comparison to the MDA-MB-231 group (PDX: <1%; MDA-MB-231: 32.2% ± 6.1 CD45^+^). Interestingly, the CD8 but not the CD4 T cells expressed high levels of TIM-3 in the MDA-MB-231 tumors. By contrast, more LAG-3^+^ cells were found in the CD4 than in the CD8 T cells in the tumor (Figure 3A). Apart from that, elevated levels of PD-1^+^ TIM-3^+^ and/or LAG-3^+^ CD4 and CD8 T cells were observed in the lymph nodes and the spleen compared to the cells found in the blood, also in the PDX mice (Figure 3A).

Next, we focused on the immune checkpoint profile of PD-1, TIM-3, and LAG-3 and applied a t-SNE analysis. PD-1 was co-expressed with TIM-3 and LAG-3 on the T cells in all the mice analyzed resulting in a triple-positive phenotype (one representative mouse is given in Figure 3B,C). Further characterization of the LAG-3-expressing T cells revealed a subpopulation of LAG-3^+^, CD49b^+^, and CD45RA^−^, established as the immunosuppressive TR-1 cells (Appendix A).

### 3.5. Correlation Analysis between Candidates for Concomitant Checkpoint Blockade Confirms PD-1, TIM-3 and LAG-3 as Targets in Breast Cancer Patients

To correlate our findings derived from the HTM with patient data, we analyzed a dataset containing 3039 primary breast cancer samples on the R2 Genomics Analysis and Visualization Platform (Figure 4, “R2: Tumor Breast (primary)−Gruvberger-Saal−3207−tpm−gse202203”).

The analysis revealed that the PD-1, LAG-3 and TIM-3 gene expression differed between the breast cancer subtypes (Figure 4A). In line with this, the quantified concentrations of soluble proteins in HTM (Figure 2A) mirrored the pattern found in the dataset (Figure 4A). Especially, (s)TIM-3 was strongly increased in comparison to the other checkpoint molecules, not only in the TNBC but also in the other breast cancer subtypes, which was confirmed by the dataset. In addition, the analyses demonstrated a positive correlation between *PDCD1* (PD-1) expression and *LAG3* (LAG-3) as well as *HAVCR2* (TIM-3) (Figure 4B, Table 2). *PDCD1* showed even a strong correlation with *LAG3* in all four subtypes (Table 2). The correlation between *PDCD1* and *HAVCR2* was only weak to moderate.

Increased expression of the PD-1 and LAG-3 genes was significantly associated with improved overall survival (OS) in patients with TNBC and ER^−^/HER2^+^ tumors but not with ER^+^/HER2^+/−^ tumors (Appendix A). There was no significant difference between the patients with high or low TIM-3 gene expression, except in the ER^+^/HER2^+^ group, in which elevated TIM-3 expression levels were associated with a worse prognosis.

## 4. Discussion

Several checkpoint proteins, such as CTLA-4, PD-1, TIM-3 and LAG-3, have been identified as regulatory molecules suppressing an effective patient-inherent immune defense of cancer, mainly by T cells. However, the relation between the expression and secretion of these molecules and their complimentary function is still largely unknown.

Different checkpoint-inhibiting antibodies against PD-1 or PD-L1 have been tested in humanized mouse models, showing inhibition of tumor growth and T cell activation in different tumor entities [23,24,25,26,27,28,29]. However, the clinical success rates using blocking antibodies against those checkpoints need to be improved.

Here, we describe the expression and the secretion of different checkpoint proteins (e.g., PD-1, PD-L1, PD-L2, galectin-9, TIM-3, and LAG-3) in HER2^+^, hormone receptor-positive and TNBC-based HTM. This is the first description of the profile of soluble immune checkpoint proteins in humanized mice. Our findings indicate that TIM-3 and its binding partner galectin-9 represent alternative (or additional) targets in breast cancer patients. The unique power of HTM lies in the coexistence of human tumor growth and a functional human immune system. Thus, this preclinical in vivo model is particularly suitable to investigate checkpoint blockade and the combinatory strategies thereof. Elucidating whether soluble factors might serve as prognostic or predictive biomarkers for the treatment response could contribute to the success of immunotherapies in cancer patients. Monitoring these molecules during the course of therapy might help assess the success of the chosen therapy, could identify resistance mechanisms, or would allow for the adaptation therapy in case of tumor progression.

Above all, the MDA-MB-231 cells (TNBC) secreted a broad range of cytokines and other soluble proteins under cell culture conditions. This might explain the strong rejection capacity of tumor cells in humanized mice. High amounts of the cytokine IL-6 were released by the MDA-MB-231 cells. Even though IL-6 is considered a pro-inflammatory cytokine, it has been investigated as a target in breast cancer, namely for the estrogen receptor modulator bazedoxifene. The experimental treatment of the TNBC revealed reduced viability and the successful blockade of proliferation and migration [30]. A similar discrepancy can be found for the highly secreted CX3CL1 protein, also known as fractalkine. This chemokine ligand chemoattracts and recruits immune cells as T cells or monocytes, but it was also already described as pro-tumorigenic and pro-metastatic [31]. Moreover, high levels of CXCL8 (IL-8) were found. CXCL8 attracts and stimulates immune cells, but it was also shown to promote angiogenesis and to trigger tumor cell proliferation and progression [32]. These controversial findings concerning pro-inflammatory and simultaneously pro-tumorigenic function require further elucidation.

Analysis of the mouse serum showed that the secretion of several checkpoint molecules, such as sTIM-3, sPD-L2 and galectin-9, was elevated, in particular in the MDA-MB-231 model, followed by the JIMT-1 (HER2^+^) transplanted mice. In line with this observation, increased numbers of tumor infiltrating lymphocytes (TILs) have been found in TNBC and HER2^+^ human breast cancers [33,34]. These two subentities are considered to represent breast cancer with enhanced immunogenicity compared to hormone receptor-positive variants. Despite this, TNBC has a poor prognosis due to the limited treatment options. However, pronounced immune cell infiltration has been associated with better OS in TNBC and also in HER2^+^ patients [35,36].

The soluble CTLA-4 levels were low, which might reflect the clinical situation, as anti-CTLA-4 immunotherapy is not yet suitable for the treatment of breast cancer patients. Although anti-PD-(L)1 therapy is applied in breast cancer patients, the levels of sPD-L1 were only marginally higher. The soluble form of LAG-3 was detectable only in small amounts in the HTM, although elevated LAG-3 expression was found on the CD4 and CD8 T cells in the MDA-MB-231 HTM and PDX model as well as in the dataset of breast cancer patient samples. The cleavage of LAG-3 is mediated by two transmembrane metalloproteases, ADAM-10 and ADAM-17, which are also responsible for the cleavage of PD-L1 [37]. It is assumed that the soluble forms can retain their ability to bind their corresponding partners. For instance, sPD-L1 has been shown to trigger similar signaling pathways to membrane-bound PD-L1 [38,39]. However, the functionality of sLAG-3 is being controversially discussed [40]. sLAG-3 is able to activate and induce maturation of antigen-presenting cells through MHC II, which in turn could contribute to the corresponding T cell responses. However, DCs derived from monocytes in the presence of sLAG-3 exhibit impaired function for antigen presentation [41].

Several soluble checkpoint molecules have been found in the serum of patients. However, not only is the number of publications that have addressed soluble checkpoints in breast cancer (patients) limited, but also correlation analyses between expression and secretion are lacking. Accordingly, the prognostic or predictive values are still uncertain. Only a few clinical studies have analyzed soluble checkpoints in the serum of early breast cancer patients. A reduction in soluble stimulatory as well as inhibitory immune checkpoints (e.g., PD-L1, CTLA-4, TIM-3) was observed in the patients compared to healthy donors [42]. Others reported the positive prognostic impact of sLAG-3 in hormone receptor-positive breast cancer [43]. In contrast, membrane-bound LAG-3 in the residual tissues, especially in combination with PD-L1 expression, has been associated with poor prognosis in TNBC [44].

Considering sLAG-3 in malignancies beyond breast cancer, high sLAG-3 levels were related to poor outcomes in head and neck cancer and hepatocellular carcinoma patients [45,46]. Moreover, Machiraju and colleagues found an association between high pre-treatment serum levels of sLAG-3 (but not sPD-1, sPD-L1 or sTIM-3) and resistance to anti-PD-1 therapy in melanoma patients [47]. However, similar to the results concerning hormone receptor-positive breast cancer, high levels of sLAG-3 have been correlated with improved patient survival in gastric cancer [48].

Regarding the immune cell-associated checkpoints, we identified a significant number of cells simultaneously expressing PD-1, TIM-3, and LAG-3. The presence of these triple-positive cells might be an indicator for combined immunotherapeutic treatments against (breast) cancer. Previous expression profiling revealed an upregulation of LAG-3 in breast cancer patients, predominantly in the TNBC, HER2^+^, and luminal A subtypes but not in the luminal B subtypes. In addition, LAG-3 expression was closely related to an enhanced malignancy of breast cancer and poor clinico-pathological factors [49]. Interestingly, we found in the HTMs a weak correlation between sLAG-3 and sPD-1 but not between sTIM-3 and sLAG-3 or sPD-1. Likewise, a strong PD-1 and LAG-3 relation in the breast cancer patients was observed. Burugu and colleagues also described the co-expression of LAG-3 and PD-1 and detected LAG-3^+^ TILs, especially in hormone receptor-negative breast cancers, which signified LAG-3 as an independent favorable prognostic factor [33]. This favorable association was also reported by other authors, in particular in the context of the ER^−^, HER2^+^, and basal-like subtype [33,50]. Bottai and colleagues also described concurrent PD-1 and LAG-3 expression on TILs in 15% of the rather aggressive TNBC subtype [51]. However, they did not find significantly improved survival in patients with tumors harboring LAG-3^+^ cells. These inconsistent results indicate a complex relationship between LAG-3 expression, prognosis, and therapy response that requires both intensified preclinical and clinical research. Furthermore, LAG-3 displays an interesting molecule regarding the so-called type 1 regulatory T cells. TR-1 cells are characterized by strong immunosuppressive function and simultaneous LAG-3 and CD49b expression [7], and they were also present in the HTM.

TIM-3 expression has been associated with shorter OS in various cancers, including cervical, lung and urothelial cancer, renal cell carcinoma, and others [52]. Comparable to the levels of sTIM-3, galectin-9 (which displays the corresponding partner of TIM-3) was also elevated in the HTM. Galectin-9 has been described to enhance the expansion of myeloid-derived suppressor cells (MDSCs) in murine models [53]. In addition, it can interact not only with TIM-3 but also with PD-1. Therefore, a blockade of galectin-9 triggers expansion of intra-tumoral cytotoxic T cells [54].

However, the prognostic value of galectin-9 in tumors has been controversially discussed. Some studies described better outcomes for patients with various solid tumors, such as breast cancer [55], melanoma [56], hepatocellular carcinoma [57], colon cancer [58], and bladder cancer [59]. In contrast, others found that galectin-9 is associated with worse clinical outcomes in patients with breast cancer [60], renal cancer [61], gastric cancer [62], and lung cancer [63]. Interestingly, there was a significant correlation between sTIM-3 and galectin-9 in our HTM. Moreover, the TIM-3 and galectin-9 gene expression was strongly elevated in all the subtypes.

A poor prognosis has been repeatedly attributed to high sPD-L1 levels in various cancer types [64]. Accordingly, the sPD-L1 plasma levels were found to be significantly higher in patients with advanced breast cancer than in patients with early breast cancer [65]. Likewise, the presence of PD-L2 was also associated with a poor prognosis in hepatocellular carcinoma, renal cancer [66], and hormone receptor-positive breast cancer [67]. High levels of PD-L2 were identified on extracellular vesicles, which were associated with tumor progression and reduced OS in TNBC [68]. Moreover, a treatment-related study revealed the association of sPD-L2 with platinum resistance in ovarian cancer [69]. Here, we found a correlation of PD-L1 on the tumor cells with sPD-1 and sPD-L1 but not sPD-L2. The expression of MHC II on the tumor cells was not connected to the amount of sLAG-3. Not surprisingly, the expression of PD-1 on the immune cells correlated with the sPD-1 found in the serum.

Overall, the prognostic and/or predictive value of checkpoint expressions, and especially the secreted variants and their mechanism of action during checkpoint blockade therapy, remain largely unclear. The Kaplan–Meier estimation curves from our database analysis revealed the significantly reduced OS of the ER^−^/HER2^+^ and TNBC patients when suffering from malignancies with low *PDCD1* and *LAG3* expression. Lower amounts of *LAG3* and *HAVCR2* seem to have a positive effect on the OS of patients with ER^+^/HER2^+^ tumors. This finding underlines the complex and in particular different impacts of immune checkpoints in individual breast cancer subtypes.

The flow cytometry data as well as the soluble factors detected in the MCF-7 HTM confirm a less aggressive phenotype. The JIMT-1 and MDA-MB-231 cells represent less differentiated breast cancer subtypes secreting multiple factors (MDA-MB-231 > JIMT-1). This could contribute to the higher immunogenicity in MDA-MB-231 that is also observed in the corresponding TNBC. In line with this, checkpoint therapies tend to be more efficient in these two subtypes than in luminal breast cancer. Nonetheless, several clinical trials are ongoing targeting additional checkpoints besides the PD-(L)1 axis, also including breast cancer patients [70]. Especially, a potential LAG-3 blockade is being evaluated in nearly 100 clinical trials. Mechanistically different approaches are ongoing, and monoclonal antibodies (e.g., IMP701, REGN3767), LAG-3-Ig fusion proteins (IMP321) or LAG-3 bispecific therapies (e.g., IB1323, CB213) targeting PD-1 and LAG-3 (MGD013, EMB-02), or CTLA-4/LAG-3 (XmAb841) are the subject of studies [71].

The intensified interest in LAG-3 combination strategies might be due to the persuasive results of the global, randomized, phase 2–3 study (RELATIVITY-047) in melanoma patients that combined the application of relatlimab (anti-LAG-3) and nivolumab (anti-PD-1). This novel dual-checkpoint therapy has just recently (March 2022) been approved by the FDA for the treatment of unresectable or metastatic melanoma. In gynecological oncology, this combined anti-PD-1 and anti-LAG-3 (REGN3767) antibody approach is under clinical investigation for the treatment of early-stage high-risk HER2^−^ breast cancer patients (NCT01042379). The LAG-3 fusion protein has been also tested for hormone receptor-positive metastatic breast carcinoma patients (NCT02614833). Another clinical study using the bispecific PD-1 and LAG-3 protein includes unrespectable or metastatic neoplasms tumors, including HER2^+^ advanced cancer and TNBC (NCT03219268). The first results indicate an acceptable safety profile and evidence of an anti-tumor response [72].

Moreover, besides the LAG-3 blockade, TIM-3-targeting strategies have been developed, including monoclonal antibodies, combination therapies and bispecific antibodies (e.g., against TIM-3 and PD-1, (NCT04785820) and are the subject of clinical trials [52]. In addition, breast cancer patients have participated in clinical trials analyzing sabatolimab (MBG453, anti-TIM-3 mAb) as a single agent or in combination with spartalizumab (anti-PD-1 mAb) [73].

All these trials are indicative of the expected potential of advanced (e.g., combined) immunotherapies. The upcoming results will determine the next generation of immunotherapeutics. A deeper knowledge about checkpoint interaction, secretion and the identification of predictive markers is definitely essential for successful implementation in clinical practice.

## 5. Conclusions

Humanized tumor mice have been proven to be a suitable patient-like model, in particular for the investigation of immunomodulatory therapies. Here, we describe not only the expression of different checkpoint molecules but also the secretion of their soluble factors. TIM-3, galectin-9, LAG-3 and PD-L2 are expressed in patient-derived breast cancer tissues and are continuously expressed and secreted by different breast cancer-based HTM. Both the cellular expression of immune checkpoint molecules and their secretion give cause to extensively evaluate dual-checkpoint blockade strategies for the treatment of breast cancer and other malignancies. HTM represent a powerful preclinical translational in vivo model to examine targeting strategies specifically tailored to checkpoint expression and secretion. Moreover, HTM allow for the preclinical assessment of treatment efficiencies and potential side effects before transferring individualized treatment approaches to clinical use.

## Figures and Tables

**Figure 1 cancers-15-02615-f001:**
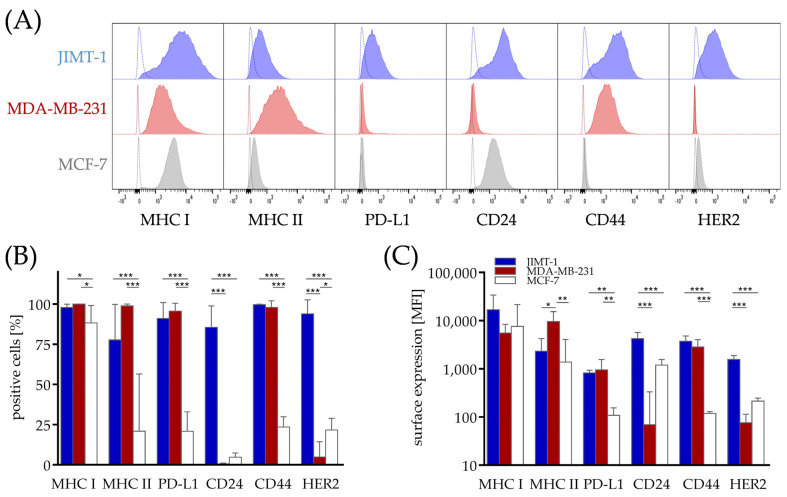
Differences in the surface marker profile associated with the immune cell response and metastasizing capacity in breast cancer cell lines from different subtypes. JIMT-1, MDA-MB-231 and MCF-7 breast cancer cells were transplanted orthotopically into humanized NSG mice. Five weeks after tumors were palpable, the tumors were harvested, processed to a single-cell suspension and the cells were subsequently analyzed via flow cytometry regarding their MHC I, MHC II, PD-L1, CD24, CD44 and HER2 expression in the JIMT-1 (filled, blue), MDA-MB-231 (filled, red) and MCF-7 cells (filled, grey). (**A**) Representative flow cytometry blots show the surface marker expression (filled histograms) with isotype staining for each measurement (open histograms). (**B**) Positive cells were determined on the basis of the isotype controls and (**C**) the associated MFI are shown as the mean ± SD for JIMT-1 (blue, *n* = 7), MDA-MB-231 (red, *n* = 5) and MCF-7 (white, *n* = 6), as compared by one-way ANOVA, Tukey’s multiple comparisons test, * *p* < 0.05, ** *p* < 0.01, *** *p* < 0.001. MFI−median fluorescence intensity, MHC−major histocompatibility complex, PD-L1−programmed death 1 ligand 1.

**Figure 2 cancers-15-02615-f002:**
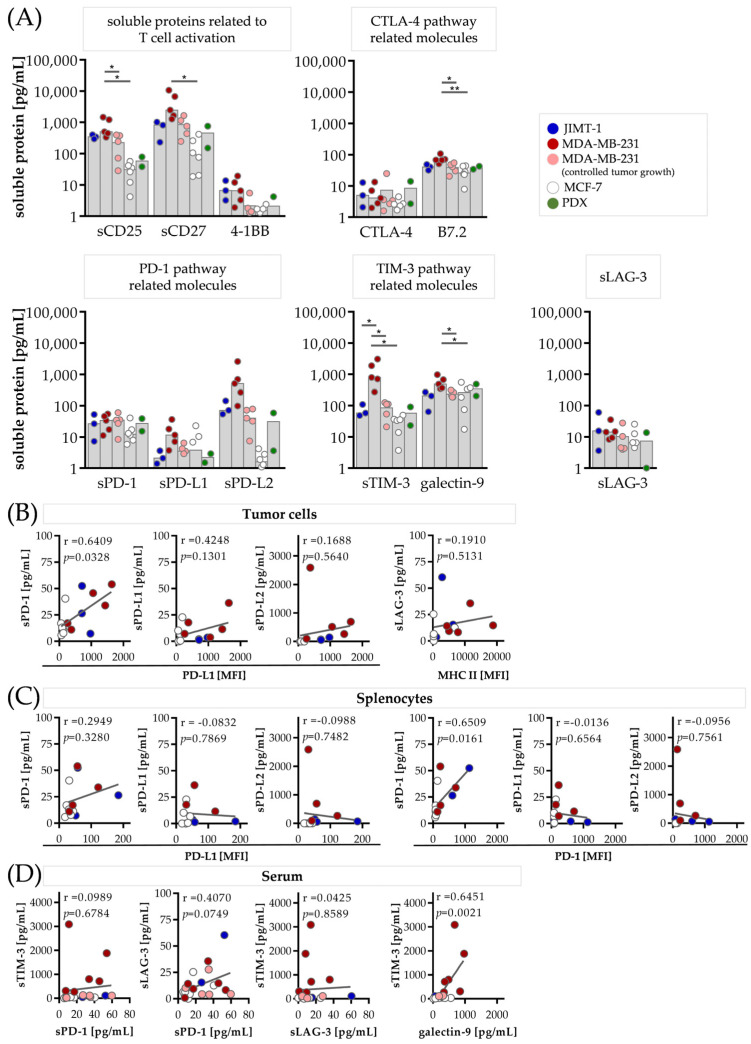
Soluble proteins involved in the immune response and immune checkpoint pathways in the serum of the HTMs. JIMT-1, MDA-MB-231, MCF-7 breast cancer cells and PDX tissue were transplanted orthotopically into humanized NSG mice. Five weeks after tumors were palpable, the tumors, spleens and serum were collected. The tumors and spleens were processed to a single-cell suspension and the cells were subsequently analyzed via flow cytometry (JIMT-1: in blue, *n* = 3; MDA-MB-231: mice with pronounced tumor growth, defined by tumor weight > 0.2 g in dark red, *n* = 6, and mice with controlled tumor growth in light red, *n* = 5, defined by tumor weight < 0.1 g; MCF-7: in white, *n* = 6; PDX: in green, *n* = 2). (**A**) The median of the soluble factors in the murine serum of different proteins involved in the immune response was analyzed via bead-based immunoassays using flow cytometry, including human sCD25, sCD27, 4-1BB, CTLA-4, B7.2, sPD-1, sPD-L1, sPD-L2, sTIM-3, galectin-9, and sLAG-3. JIMT-1, MDA-MB-231 (both groups) and MCF-7 were compared via one-way ANOVA, Tukey’s multiple comparisons test, * *p* < 0.05, ** *p* < 0.01. (**B**–**D**) Correlations between the various soluble molecules and membrane-bound proteins on the tumor cells and splenocytes isolated from the JIMT-1, MDA-MB-231 and MCF-7 HTMs are shown. Correlations between (**B**) sPD-1, sPD-L1, sPD-L2 and the MFI of membrane-bound PD-L1 as well as between sLAG-3 and the MFI of membrane-bound MHC II on the tumor cells, (**C**) between sPD-1, sPD-L1, sPD-L2 and the MFI of membrane-bound PD-L1 and PD-1 on the splenocytes and (**D**) between sPD-1, sLAG-3, sTIM-3 and sTIM-3 and galectin-9 in the serum are displayed.

**Figure 3 cancers-15-02615-f003:**
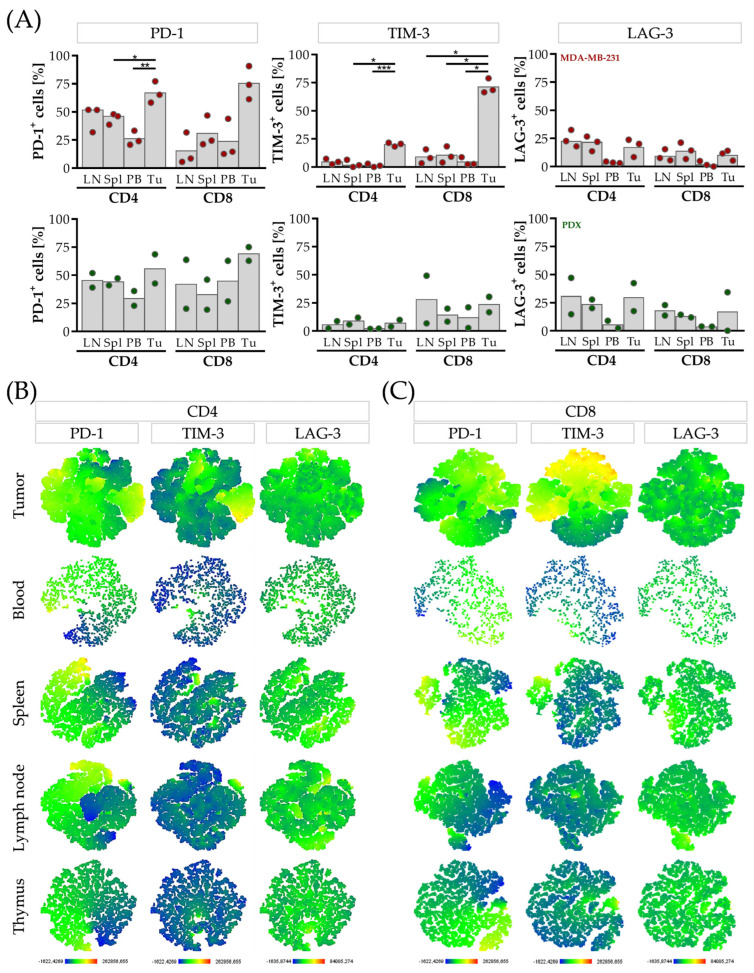
(Co-)expression of the checkpoint molecules PD-1, TIM-3 and LAG-3 on the CD4 and CD8 T cells derived from different organs in the humanized mice. MDA-MB-231 cells and PDX tissue were transplanted orthotopically into humanized NSG mice. Five weeks after tumors were palpable, the blood, spleens, lymph nodes, thymi and tumors were collected. The organs were processed to a single-cell suspension and the T cells (CD45^+^, CD3^+^, CD4^+^ and CD45^+^, CD3^+^, CD8^+^, respectively) were analyzed via flow cytometry regarding their PD-1, TIM-3 and LAG-3 expression. The T cells derived from the MDA-MB-231 mice are depicted in red, the T cells derived from the PDX in green. (**A**) Percentages of PD-1^+^, TIM-3^+^ and LAG-3^+^ cells among the CD4 and CD8 T cells in the lymph nodes (LN), the spleen (Spl), in peripheral blood (PB) and in the tumor (Tu) are shown. Each symbol represents one individual mouse (median). The MDA-MB-231 cells were compared via one-way ANOVA, Tukey’s multiple comparisons test, * *p* < 0.05, ** *p* < 0.01, *** *p* < 0.001. (**B**) CD4 and (**C**) CD8 T cells derived from different organs were analyzed using t-SNE maps regarding their PD-1, TIM-3 and LAG-3 co-expression, color-coded by the expression as indicated. One representative MDA-MB-231 HTM is shown.

**Figure 4 cancers-15-02615-f004:**
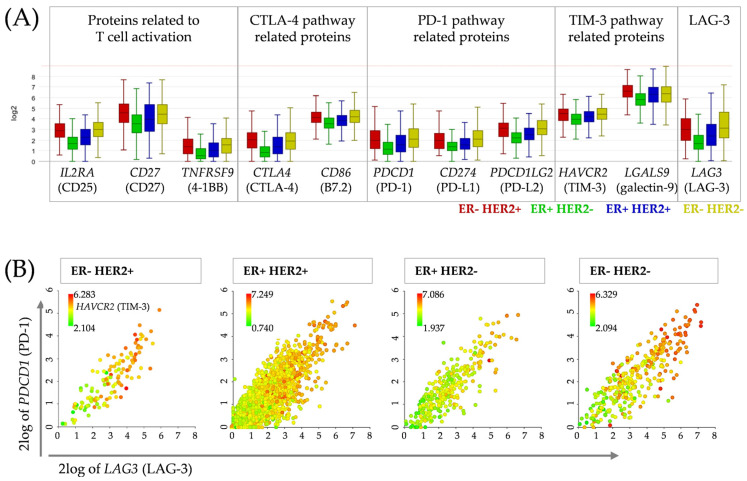
Gene expression and correlation of genes involved in the immune response and immune checkpoint pathways in breast cancer patients. A dataset containing 3039 breast tumors (“R2: Tumor Breast (primary)−Gruvberger-Saal−3207−tpm−gse202203”) was analyzed. (**A**) Gene expression of CD25 (*IL2RA*), CD27 (*CD27*), 4-1BB (*TNFRSF9*), CTLA-4 (*CTLA4*), B7.2 (*CD86*), PD-1 (*PDCD1*), PD-L1 (*CD274*), PD-L2 (*PDCD1LG2*), TIM-3 (*HAVCR2*), galectin-9 (*LGALS9*), and LAG-3 (*LAG3*) was compared between the four clinical groups: ER^−^ HER2^+^ (red, *n* = 124), ER^+^ HER2^−^ (green, *n* = 2308), ER^+^ HER2^+^ (blue, *n* = 287) and ER^−^ HER2^−^ (yellow, *n* = 320). (**B**) Correlation of *LAG3* and *PDCD1* gene expression, with additional color-coded *HAVCR2* expression, is shown. Patient samples were ordered by the *PDCD1* and *LAG3* levels.

**Table 1 cancers-15-02615-t001:** Cytokines and other soluble factors associated with immunogenicity secreted by breast cancer cell lines derived from different subtypes. JIMT-1, MDA-MB-231 and MCF-7 breast cancer cells were cultured for 72 h under normal cell culture conditions (5% FCS). After medium exchange, the cells were cultured under starving conditions with 1% FCS and the supernatants were collected after additional 48 h. HMGB1 and CX3CL1 (=fractalkine) secretion was analyzed via ELISA. The human IL-6, TGF-ß, IL-4, IL-1ß, TNF, CXCL8 (=IL-8), MCP-1 (=CCL2), CXCL10 (=IP-10), galectin-9, sPD-L1 and sPD-L2 concentrations were analyzed via bead-based immunoassays using flow cytometry. The mean ± SD in pg/mL is shown (*n* = 4).

	Cytokines and Ligands for Chemokines										Ligands for Checkpoint Proteins			
	IL-6	TGF-ß	IL-4	IL-1ß	TNF	CXCL8	MCP-1	CXCL10	CX3CL1		HMGB1	Gal-9	sPD-L1	sPD-L2
JIMT-1	214	(-)	(-)	(-)	(-)	14	(-)	(-)	(-)		44,578	(-)	(-)	257
	±82					±12					±12,795			±106
MDA-MB-231	5672	44	(-)	(-)	(-)	1194	120	71	6751		52,257	59	(-)	206
	±1237	±32				±110	±36	±13	±729		±7768	±36		±69
MCF-7	(-)	(-)	(-)	(-)	(-)	(-)	(-)	(-)	(-)		32,056	(-)	(-)	(-)
											±18,210			

PD-L1/2−programmed death 1 ligand 1/2, IL−interleukin, TGF-ß−tumor growth factor ß, TNF−tumor necrosis factor, CXCL−chemokine (C-X-C motif) ligand, MCP-1−monocyte chemoattractant protein 1, HMGB1−high mobility group box 1 protein, Gal-9−galectin-9.

**Table 2 cancers-15-02615-t002:** Correlation analysis between several possible candidates for concomitant checkpoint blockade for immunotherapy in breast cancer patients. A dataset containing 3039 breast tumors (“R2: Tumor Breast (primary)—Gruvberger-Saal—3207—tpm—gse202203”) was analyzed. Pearson’s correlation was calculated. All the correlation coefficients indicate high significance. Color code displays the strength of the relationship.

	Correlation Coefficient			
	ER^−^ HER2^+^	ER^+^ HER2^+^	ER^+^ HER2^−^	ER^−^ HER2^−^
*PDCD1* vs. *LAG3*	0.865	0.850	0.777	0.835
*PDCD1* vs. *HAVCR2*	0.495	0.474	0.433	0.622
*LAG3* vs. *HAVCR2*	0.590	0.549	0.490	0.635
*LAG3* vs. *LGALS9*	0.567	0.659	0.587	0.706

## Data Availability

The following are available online at http://www.mdpi.com/ethics (accessed on 25 July 2018).

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
