# Peer review of "Immune Checkpoint Profiling in Humanized Breast Cancer Mice Revealed Cell-Specific LAG-3/PD-1/TIM-3 Co-Expression and Elevated PD-1/TIM-3 Secretion"

_cancers, 2023, doi:10.3390/cancers15092615_

Round 1

Reviewer 1 Report

Immune checkpoint profiling in humanized breast cancer mice 2 revealed cell-specific LAG-3/PD-1/TIM-3 co-expression and elevated PD-1/TIM-3 secretion Bruss etal is interesting, however it needs additional data to strengthen the manuscript.

Comments

1.     Fig 1. The labels are very small, the comparison and p values are missing.

2.     Fig 2. The labels are very small, the comparison and p values are missing.

3.     Most figures are showing only mRNA expression, it will be nice to see the protein blots along with mRNAs.

Author Response

please find our response attached

Reviewer 2 Report

An appropriate preclinical mouse model is needed to investigate new immunotherapies, and the authors describe the expression of various immune checkpoint molecules and soluble factors in three different humanized models of breast cancer. They identified TIM-3 and potentially LAG-3 as important targets and/or biomarkers in breast cancer using humanized mouse models and analysis of breast cancer samples. Although the approach is potentially important, several important weaknesses limit the conclusions of this manuscript. First, the data are correlative and need to be validated using pharmacological and/or genetic approaches. The small sample sizes (n=3) for the JIMT-1 model in Figures 1-2 limits statistical power and the ability to show differences between groups. The small sample sizes also make the conclusion that PD-1, LAG-3 and TIM-3 expression differ between breast cancer subtypes (Fig 4A) less convincing.

Author Response

Please find our response attached

Reviewer 3 Report

In the submitted manuscript authors conducted an immune checkpoint profiling in humanized breast cancer mice models and observed cell-specific LAG-3/PD-1/TIM-3 co-expression and elevated PD-1/TIM-3 secretion, concluding that LAG-3 and TIM-3 represent additional key molecules within the breast cancer anti-immunity landscape.

Although very robust and quite well written, with lots of results corroborating authors' hypotheses, this manuscript has several drawbacks which must be corrected and further improved before this manuscript is suitable for publication.

1) Authors should inspect https://www.genenames.org/ and https://www.uniprot.org/ and uniformly use only approved gene/protein names and symbols, with clear differentiation between genes and proteins. Also, there is no need to capitalize each word of gene/protein full name.

2) A figure depicting immune checkpoints mentioned in 'Introduction' would help in understanding the text.

3) All used abbreviations should be explained in both 'Abstract' and main text after first mentioning (e.g., TNBC, HTM, mab, PDX, SD, etc.).

4) Exact model and manufacturer of a plate reader used for ELISA should be mentioned. The same is for all used instruments.

5) Title "2.8. Database analyses" is misleading since you analyzed only one dataset using only one web-based tool.

6) Section "2.9. Statistical analyses" is obviously redundant since you haven't performed any statistical analyses besides correlation analyses, which wasn't even mentioned! Differences between results obtained with various HTM models (cell-based or PDX) should be statistically inferred, not just ad hoc concluded based on absolute number values! Also, EVERY corr. coeff. and its p-value must be provided also in the main text and figures, not just those (presumably) stat. significant. Also explain how you categorized extent of correlation (moderate, weak, etc.). There is also no need to constantly write "SD" after stating SD value.

7) Information on breast cancer tissue samples used for PDX must be provided.

8) Re-check section numeration because there are two 2.2 and 3.3 sections.

9) t-SNE analysis must be explained in 'Methods'.

10) Text on Figure 2B-D and Figure 4B is too small and thus unreadable. P-value must always be provided with corr. coeff. The same is with Table 2, or it must be stated that only stat. significant corr. coeff. were presented in Table 2.

11) Writing style of 'Results' is more similar to 'Discussion' so it is not very easy to follow.

12) In 'Discussion' authors wrote a lot about prognostic significance. In that light, they could use R2 Genomics Analysis and Visualization Platform and TCGA-BRCA or some other BC datasets to inspect prognostic significance of studied immune checkpoints.

Author Response

please find our response attached

Round 2

Reviewer 1 Report

The quality of the manuscript is improved

Reviewer 2 Report

Authors have adequately addressed the prior comments from reviewers.

Reviewer 3 Report

Authors have satisfactorily responded to all my concerns and substantially improved quality of this manuscript.